# Flexible Pressure Sensors for Integration into Karate Body Protector

**DOI:** 10.3390/s23146524

**Published:** 2023-07-19

**Authors:** Derya Tama Birkocak, Pedro Gomes, Helder Carvalho

**Affiliations:** 12C2T—Centre for Textile Science and Technology, University of Minho, 4800-058 Guimarães, Portugal; derya.tama@ege.edu.tr; 2Textile Engineering Department, Ege University, 35040 İzmir, Türkiye; 3Textile Engineering Department, University of Minho, 4800-058 Guimarães, Portugal; vpcgomes@gmail.com

**Keywords:** piezoresistive pressure sensors, textile sensors, karate sport, smart body protector

## Abstract

The increasing interest in karate has also attracted the attention of researchers, especially in combining the equipment used by practitioners with technology to prevent injuries, improve technical skills and provide appropriate scoring. Contrary to the sport of taekwondo, the development of a smart body protector in the sport of karate is still a niche field to be researched. This study focused on developing piezoresistive, textile-based pressure sensors using piezoresistive film, conductive fabric as well as different bonding materials and methods. Primarily, small-scale sensors were produced using ultrasonic welding, hot press welding and oven curing. These were characterized using a universal testing machine and specific conditioning and data-acquisition hardware combined with custom processing software. Large-scale sensors were then manufactured to be placed inside the karate body protector and characterized using cyclic testing. The conditioning circuit allows flexible gain adjustment, and it was possible to obtain a stable signal with an output of up to 0.03 V/N, an adequate signal for the tested force range. The transfer function shows some drift over the cycles, in addition to the expected hysteresis and slight nonlinearity, which can be compensated for. Finally, the configuration with the best results was tested in real practice tests; during these tests the body protector was placed on a dummy as well as on a person. The results showed that the piezoresistive textile-based pressure sensor produced is able to detect and quantify the impact of even light punches, providing an unobtrusive means for performance monitoring and score calculation for competitive practice of this sport.

## 1. Introduction and Objectives

Flexible, thin, lightweight pressure sensors are an interesting component for electronic wearables, with the potential for use in several relevant applications. They can be used in monitoring of bed-rest patients, insole pressure measurement in shoes, optimization of mattress design, force measurement in sports and other applications.

Smart textiles, which attract great attention due to their high-added-value features and allowing the use of advanced technology, can be defined as systems that are able to sense and respond to environmental changes by detecting the condition of the wearer and processing this information [1,2,3,4,5]. The increasing interest in wellness by users has made consumers’ preferences change from being comfortable to the use of wearable smart devices. Still, in addition to being functional, wearable electronics are required to be light-weight as well as not restrictive of the movements of the user. Herewith, the importance of the use of textile-based sensors comes to the forefront due to their flexible, light and comfortable features.

Many researchers have studied the development of textile sensors, which are categorized as strain sensors [6,7,8,9], pressure sensors [10,11,12,13,14,15], temperature sensors [16] and humidity sensors [17,18,19]. Moreover, the textile pressure sensors, which are also the subject of this study, have different working principles, namely, piezoresistive [12,13,14,20,21], piezoelectric [22,23,24,25,26], capacitive [10,27,28,29,30,31] and optical [32]. The present study focused on developing piezoresistive material-based textile pressure sensors.

Piezoresistive materials are materials that undergo a change in electrical resistance when pressure is applied [2]. Due to this feature, they are used as a common conversion means for micro-electromechanical systems, especially force sensors, pressure sensors, strain sensors, microphones, accelerometers, temperature sensors and chemical sensors. Piezoresistive pressure sensors can be produced in two constructions: the sandwich layout (Figure 1—left) and the single-layer structure (Figure 1—right) [21]. In sandwich sensors, there is a piezoresistive material in the middle layer. Electrodes are placed on both sides of this material using conductive materials and covered with an insulating material to prevent short circuits. In the single-layer structure, the electrodes are positioned on only one side of the piezoresistive material so that they do not touch each other.

Various materials are used as piezoresistive layers and electrodes in piezoresistive material-based pressure sensors. Piezoresistive conductive polymer composites are widely used fillers for their remarkable chemical stability as well as their good electronic and mechanical features [33]. Therefore, piezoresistive carbon-filled polyethylene plastic film was used in this study, which is also commonly preferred due to its low cost, simplicity to process, good resolution and flexibility [34]. Different materials are applied as electrodes such as metal materials, carbon nanomaterials, ionic hydrogel and conducting polymers; an ideal stretchable electrode should maintain high conductivity under large strains, considering the usage of such electrodes in pressure sensors [3,33]. Especially, fabric electrodes fabricated using these materials are attracting attention due to their thermal behavior and washability [34,35,36,37]. Several authors have studied the fabrication of piezoresistive material-based flexible pressure sensors. 

Luo and Liu (2013) fabricated a single-wall carbon nanotube (SWCNT) thin-film piezoresistive sensor using the spray-coating technique and found out that the film thickness and the microstructure of the SWCNTs had effects on the sensor’s piezoresistive behavior, where the tensile strain was between 20–30% [38]. In the small tensile strain range of 0–2%, the effects of the film thickness and the microstructure of the SWCNTs were negligible. Muller et al. (2015) developed a pressure sensor using piezoresistive film [39]. In their study, the electrodes were installed with strips of copper-coated fabric oriented perpendicular to the lower layer, forming a matrix. Zhu et al. (2018) manufactured a sensor based on the piezoresistive principle utilizing force sensors using flexible piezoresistive PI film, by which it is possible to calculate the magnitude and direction of the 3D force applied to the film [40]. Carvalho et al. (2017) studied different electrode materials such as a conductive woven fabric, a conductive knitted fabric and copper tape [12]. The conductive woven fabric demonstrated similar behavior to that of the copper tape, with some spread and hysteresis. Binelli et al. (2023) focused on the development of additively manufactured wearable devices, specifically shoe insoles with embedded piezoresistive sensors. They used the biocompatible nature of silicone in their ink formulations. However, this is an early-stage development, and further testing and refinement are necessary [41]. Duan et al. (2023) conducted a review of research about the applications of daily carbon ink (DCI) in various sensors and suggested combining DCI with other functional materials to improve the performance of the sensors in sensing as well as their applications [42]. Gilanizadehdizaj et al. (2022) developed flexible piezoresistive pressure sensor arrays using reduced graphene oxide on flexible PCB and obtained a sensitive sensor array with a wide range as well as low hysteresis [43]. Likewise, another research team developed piezoresistive sensors as electronic skin using a reduced graphene oxide self-wrapped copper nanowire network [44]. Duan et al. (2022) published a review paper on flexible pressure sensor arrays, focusing on their various applications. They drew attention to the requirement of actual applications with flexible sensors to test their sensitivity over a wide pressure range; however, they stated that increasing the sensing range is a bigger and more important issue than sensitivity to achieve [45]. Gilanizadehdizaj et al. (2023) addressed similar challenges faced by wearable pressure sensors, such as sensitivity and pressure range; they obtained sensors that were highly sensitive at low pressures, but they lost sensitivity at higher pressure [46]. As the literature reveals, the performance of flexible pressure sensors is expected to improve, but it is essential to continue research on materials, structures, fabrication methods and integration strategies to overcome the current limitations.

In previous research works by the authors, several attempts have been made to fabricate pressure sensors based on the piezoresistive principle using different materials such as piezoresistive film, conductive silicone, conductive ink, conductive fabric, graphene nanoplatelets and various substrate materials [13,14,20,47]. As described previously, several researchers have developed flexible force sensors using various methods and materials. However, the development of large-area sensors, required for force sensing in more extensive regions, such as those necessary for force sensing in martial arts, has not been described. Moreover, this study is intended to explore possibilities of integration of these sensors into the clothing/protections normally used in these sports. 

The objective of this study is to develop a flexible piezoresistive assembly to serve as force/pressure sensor and study its integration into wearables. A practical case regarding the development of a smart body protector for sport karate is studied. In sport karate, instant observations may cause improperly awarded scores or missed points [20]. Therefore, in order to assist in proper scoring, the necessity of using sensor-integrated products during competitions has been raised. Electronic scoring/awarding systems exist for sport taekwondo, giving room for every athlete to be accountable for their successes as well as their failures. The products on the market and the existing literature [43,44,45,46] show that researchers have focused on sport taekwondo; however, research on smart body protectors in sport karate is still a niche field to be investigated. This was one of the main motivations to continue working on developing a smart body protector for sport karate. For this application, a fundamental requirement is the ability of the sensor to measure force in a large area instead of a point or limited region.

## 2. Materials and Methods

The construction of the pressure sensors built in this study consists of 3 layers (Figure 2). The piezoresistive material is placed as the middle layer, and it is overlapped by the electrodes on both sides. In order to prevent a short circuit, the electrodes were cut into smaller dimensions than the piezoresistive material to eliminate the possibility of contact with each other.

A medium-level electrically conductive polyethylene film loaded with carbon (Linqstat; Caplinq), which is generally used for large-scale and large-pressure-application pressure sensors [48], was used as the piezoresistive material. For the electrode, a silver-plated polyamide fabric (Bremen by Statex) [49] was used (Table 1).

### 2.1. Small-Scale Pressure Sensor Preparation

Three different methods were used to assemble the material layers that provide mechanical stability without affecting the electrical contact in their interactions. 

#### 2.1.1. Manufacture of the Pressure Sensors Using Ultrasonic Welding

A Pfaff 8310 Seamsonic machine was used to join the layers on the four sides by forming a square. The dimensions of the piezoresistive film were 30 mm by 30 mm, and the dimensions of the conductive fabric were 35 mm by 25 mm. The created pressure sensing area was 25 × 25 mm^2^. The machine setting was 100% ultrasound power with a sewing speed of 14 cm/min, and the roller width used was 4 mm. Figure 3 shows the layer-joining process, the construction of the sensor with the welding lines and the sensor.

#### 2.1.2. Manufacture of the Pressure Sensors Using a Hot Press 

The adhesion between layers through direct welding was studied using a hot press (Figure 4). Two processes were performed; in the first, conductive fabric layers and one piezoresistive film layer were used, while in the second, conductive fabric layers and two piezoresistive film layers were employed. In order to determine the process conditions, the melting point of the piezoresistive film was analyzed using differential scanning calorimetry (DSC), and the value obtained was 123 °C. The piezoresistive film was cut into dimensions of 25 × 25 mm^2^, and the conductive fabric was cut into dimensions of 20 × 20 mm^2^. The hot press welding process was conducted for 10 s at a temperature of 120 °C, under 3.5 bar pressure. 

#### 2.1.3. Building Pressure Sensors Using Bonding Materials

In a third approach, the adhesion between layers was provided using bonding materials. Different bonding materials were selected and placed between layers, namely, thermoplastic web (AB-Tec-A001-14g thermoplastic web—TW, Figure 5b) and thermoplastic net (Protechnic-311BD35 thermoplastic net—TN, Figure 5a). Figure 5c shows the sensors between the glass plates used in the oven.

According to the datasheets of these materials, the melting temperature ranges were 105–110 °C for TW and 78–88 °C for TN. 

Knowing these temperatures was important to determine the process conditions. The curing was performed using a lab oven for 10 min at a temperature of 110 °C. The layers were placed inside glass plates to provide a standard pressure. The sample dimensions were as follows: 30 × 30 mm^2^ for the piezoresistive film, 25 × 70 mm^2^ for the electrodes and 25 × 25 mm^2^ for the bonding materials. 

To improve adhesion, plasma surface treatment was employed on the piezoresistive material. Plasma treatment can be used to induce a cleaning effect in the samples, changing the wettability and surface texture (creating microroughness); in this way, it increases the absorption and adhesion of finishing agents, stampings and inks [50]. The generation of free radicals in the process may induce secondary reactions such as crosslinking, thus allowing for graft polymerization [51,52]. Therefore, we used plasma treatment on the Linqstat. Assuming the improvement in adhesion to be certain, according to [51,52] (which will be addressed by the contact angle measurement), it was important in this work to determine whether this treatment does or does not affect the piezoelectric properties of the material.

The amount of plasma dosage applied to the substrate is influenced by the power of the discharge, the number of passages of the fabric in the DBD (dielectric barrier discharge) application field, the width of the treatment and the speed. The total value can be calculated according to the following equation (Equation (1)) [53].
(1)$$D=\frac{P\cdot N}{s\cdot w}$$
where

*D*: Dosage [$\frac{W\cdot \min}{m^{2}}$];

*P*: Power [*W*];

*N*: Number of passages;

*s*: Speed [m/min];

*w*: Width of treatment [m] [53].

In this work, the Linqstat substrate was passed five times on each side at 4 m/min with a 1000 W discharge and a width of treatment of 0.5 m, thus resulting in a total dosage applied to each side of the sample of 2500$\;\frac{W\cdot \min}{m^{2}}$, or 41.7 $\frac{J}{m^{2}}$.

The wettability of the piezoresistive material was measured by obtaining the average contact angle before and after treatment with plasma. 

### 2.2. Large-Scale Pressure Sensor Preparation for Integrating into Body Protector

After obtaining data about the methods and used materials, the dimensions of the pressure-sensing area of the sensors were scaled up to 190 × 280 mm^2^. The large-scale sensors were prepared using the same construction, with dimensions of 280 × 320 mm^2^ for the conductive fabric and 200 × 300 mm^2^ for the piezoresistive film. The bonding materials were cut into dimensions of 190 × 280 mm^2^. Figure 6 shows the construction of the sensor prepared using TN.

In addition to using the thermoplastic net, a regular double-sided adhesive film (Silhouette America; 0.75 mm sheet) was used as a bonding material. This film allowed for exact control of the geometry of openings in the film by using a laser cutter, thus allowing us to influence the electrical resistance of the sensor and the resolution of measurement. Two double-sided adhesive films were cut with a laser cutter to form adjacent circles of 2 mm and 15 mm diameter. In Figure 7, the laser cutting process of the double-sided films, their constructions and the obtained sensors are shown. These sensors were not subjected to plasma treatment.

The film with 2 mm sized holes has an open area of 50.25% of the total, whereas the film with 15 mm sized holes has 44.16% of its area open. 

### 2.3. Pressure Sensors Testing

The sensors built in small dimensions were initially tested by a Fluke 45 multimeter in order to obtain a first evaluation of the functionality of the samples as pressure sensors. This test consists of checking whether the sensor presents a resistance value that decreases when force is applied on it. Afterwards, the functional sensors were tested using a Hounsfield dynamometer producing 10 cycles of compression between 2 and 100 N at a speed set at 5 mm/min. To provide mechanical protection and amplitude for the compression test, a layer of 3 mm EPDM (ethylene propylene diene monomer rubber) foam was placed on each side of the sensor. During the tests, the sensor was connected to a signal-conditioning circuit (Figure 8) to produce a voltage signal according to the sensor’s resistance. Equation (2) represents the relation between the sensor resistance and voltage [12].
(2)$$V_{o}=V_{off}\left(1+\frac{R}{R_{s}}\right)$$
where

*V_o_*: Voltage output;

*V_off_*: Voltage input;

*R_s_*: Resistance of sensor;

*R*: Feedback resistance [12].

**Figure 8 sensors-23-06524-f008:**
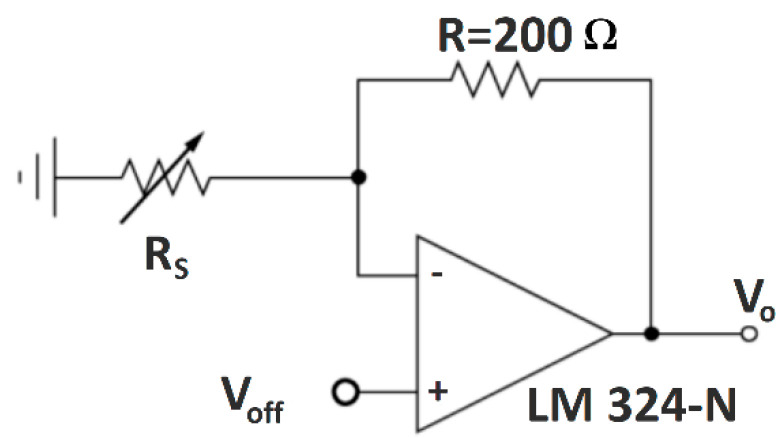
The signal-conditioning circuit for the sensors [12].

Considering that the typical relation between the force applied and the resistance of a sensor is a type of curve fitting the equation *R* = 1/F, the transfer function of this circuit has the effect of producing a more linear relation for the output (voltage) as a function of the input (force).

*V_off_* and *R* are adjusted according to the expected resistance variation of the sensor, as well as to minimize the current drawn by the circuit when resistance of the sensor is low (i.e., when the force is high).

The output voltage vs. time was acquired by a National Instruments NI-DAQ 6229 data acquisition board (DAQ) and stored in a connected laptop. The Hounsfield dynamometer provided force vs. displacement data. The force and voltage signals during the tests were synchronized using a custom-developed software in Labview. The measurement process is presented in Figure 9.

Similarly, the large-scale samples were tested using a Hounsfield universal testing machine (Figure 10) for 10 cycles at a speed of 50 mm/s, with 0 to 500 N of Force. The tests were conducted at 5 different areas on the sensor (four corners and the center).

Later on, the large-scale sensor with the best results was integrated into a karate body protector. The supplied body protector was unstitched; a pocket was sewn onto the inner layer to place the sensor into (Figure 11).

For real practice tests (Figure 12), the sensor was firstly attached to a wall while connected to the sensor signal-conditioning equipment, and punches were applied with different forces. Afterwards, the sensor was placed into the sewn pocket, and, in turn, the body protector was put on a male dummy. The testing process was repeated. In the final test, a person wore the karate body protector and made some movements: presses to the chest, deep inhalations and exhalations, jumps and simultaneous air punches. Finally, the person was punched by another person on the chest.

## 3. Results and Discussion

### 3.1. Small-Scale Sensor Results

#### 3.1.1. Sensor Assembled Using Ultrasonic Welding

The preliminary test, using a multimeter, showed that the sensor prepared with the ultrasonic welding technique was functional as a pressure sensor. The sensor was then tested with the described setup, using a feedback resistor of 200 Ω and an input voltage of 0.1 V. In Figure 13 the applied force and measured voltage signals over time are shown to provide the reader with a perception of the typical waveforms measured. These are similar in all cases.

Figure 14 shows the resulting force–voltage transfer function based on the acquired data. As can be seen, the voltage–force relation is not perfectly linear. In addition to a marked hysteresis, there is also a drift of the voltage values over the force cycles. This can be observed especially in the voltage values obtained at maximum force and is caused by the known creep behavior of this type of material, a property that has been observed by several authors and is even included by Kalantari et al. (2012) in their model for this material [54].

Moreover, the missing adhesion between layers in the central area of the sensor was found, in practice, to cause an unstable response at the zero-force point. This instability cannot be noticed in the cyclic test, given that there is always a minimum force of 2 N. It is expected that in large-area sensors this instability would be even worse because a larger contact area is looser. Therefore, no further sensors were created by this method.

#### 3.1.2. Sensors Welded Directly in the Hot Press

The samples prepared by welding in the hot press were both unfunctional as pressure sensors. The resistance was very low, on the order of 1 or 2 Ω, and did not change with pressure. This could be caused by the pressure/temperature combination in the hot press, with the silver nanoparticles contained in the conductive fabric possibly migrating into the piezoresistive film and producing a short circuit between the electrodes.

#### 3.1.3. Sensors Assembled Using Bonding Materials

Regarding the samples obtained using a bonding material between layers, the first analysis was the measurement of the contact angle. The average contact angle was measured at the Linqstat substrate, with results of 117.78° before (Figure 15a) treatment and 63.16° after treatment (Figure 15b).

Supporting the literature cited above [50,51,52,53], the results show a significant effect of the plasma treatment on the hydrophilic properties of the substrate. A subjective analysis suggests an improvement of the adhesion between layers, but the actual delamination forces were not measured; in this stage of the research, the effect of the plasma treatment on the piezoresistive properties of the sensor assembly are evaluated.

Figure 16 presents the voltage output graphs of two sensors prepared using TW as a bonding material to join the layers, with and without plasma treatment. In this case, the conditioning circuit is set up with a feedback resistance of 200 Ω, with the voltage input at 1 V.

The sensors are all functional as pressure sensors but display high nonlinearity, and drift and hysteresis are again observed. An ideal relation between voltage and force would show the obtained voltage overlapping the one measured in previous cycles. This would mean that the materials had the capacity to mechanically recover from cycle to cycle. The observed drift can be explained by the creep behavior previously mentioned. The sensor is not able to fully recover mechanically from the pressure applied in a previous cycle before the new cycle begins.

It can be seen that regarding nonlinearity, hysteresis and drift, the plasma treatment seems to have no effect, which is important to make the use of plasma treatment viable. The voltage values at maximum force seem to be higher for samples not treated with plasma (meaning that the sensors exhibit lower resistance at maximum force and are thus more sensitive). However, the differences generally observed between samples are as relevant as the difference between these two samples with and without plasma treatment. This variability is in itself an important observation and must be further studied. If plasma treatment does reduce the sensitivity of the sensor, this is easily overcome by adjusting the conditioning circuit.

For the sensor obtained using TN, the same resistance of 200 Ω was employed, but the voltage input had to be lowered to 0.1 V to adjust the output voltage to the maximum about 3.7 V of the circuit. This means that the resistance of these sensors is lower than that of the sensors with TW, which is expected, as the TN material has a larger contact area between the electrode and piezoresistive substrate. Figure 17 shows the transfer function measured for two samples.

In addition to the difference in the resistances of the sensors, there seems also to be a wider spread of the transfer functions form cycle to cycle and a larger hysteresis. This should be further quantified, but given the significant difference between the resistance values of the two samples, a larger number of sensors should be fabricated and compared. The variability of the response may be due to fabrication factors such as dimensional variations of the electrode and piezoresistive layer, the positioning of the compression foot on the sensor and others. However, there are variations in volume conductivity along the surface of the piezoresistive film due to variations in charge dispersion that are being addressed, as stated by the manufacturer himself [55,56].

Although the TW bonding material seems to perform similarly or even better than the TN material, the latter was selected for a large-scale sensor due to the regularity of its structure. This provides, in principle, a greater evenness of response over the surface of the sensor, which is important for the proposed body protector, in which it is important to have the same response at different points of the sensor. 

### 3.2. The Large-Scale Sensor Results

For the tests of the large-scale sensors, the conditioning circuit is set up with a feedback resistance of 200 Ω and an even lower value of the input voltage, 0.05 V. This is necessary because the larger area of the sensor results in a lower sensor resistance, thus producing higher output voltages.

Figure 18, Figure 19 and Figure 20 show the results of the measurement of the transfer functions presented by the sensors at different positions.

Observing the data presented in the previous three graphs, the same behavior already found in the small sensors can be identified, i.e., nonlinearity, hysteresis and drift of the values over the cycles. The drift seems to be lower when using the adhesive film instead of the TN bonding. However, due to the variability of the behavior in different positions in all three cases, this observation cannot be generalized.

It is quite evident that there is variation in the resistance of the piezoelectric film according to the location where the pressure is being applied. For the TN bonding net, which is elastic and conformable, the nonuniformity of its distribution between the electrodes and piezoresistive layer may explain some of the variability. However, the adhesive film used is not elastic or conformable, and it is perforated with great accuracy. In this case, the variability can only be explained by the presser foot applying pressure on different proportions of contact/noncontact areas, which would be most apparent in the sensor with the 15 mm holes. The other explanation is variation in the resistance of the piezoelectric film itself, which can reach 10% roll-to-roll, as specified by the manufacturer [56]. Keeping in mind that the sensor is based on volume conductivity, another factor for variation can be differences in thickness. There are alternative piezoresistive materials to be explored, with potential for more regular piezoresistive layers, such as piezoresistive inks.

It is interesting to observe that the 15 mm hole-based sensor is the one that exhibits the least variation, drift and hysteresis at positions 1.2 and 5. Conversely, the measurements at positions 3 and 4 deviate very much from the remaining measurements, both quantitatively as well as qualitatively. This is certainly due to a construction defect in the tested area. The variation of the contact area can explain the quantitative difference but not the qualitative behavior at these positions.

Considering the previous observations, the real practice tests were conducted using the sensor built with the double-sided adhesive and 2 mm holes. As mentioned before, the sensor was tested in several ways. Figure 21 presents the data obtained by placing the sensor on the wall (Figure 21a) and on the dummy (Figure 21b). Figure 22 presents the results obtained in the real practice tests. During the dummy and real practice tests, the sensor was placed inside the karate body protector.

The results show that the sensor can detect applied punches with different forces in real-use conditions (Figure 21). The sensor was sensitive enough to detect even light punches, and when a high-force punch was applied, the resistance went down, and consequently the voltage went to the highest value.

Figure 22 presents the voltage data over time obtained in the tests performed by a person who wore the pressure sensor integrated in the karate body protector. As can be clearly seen in the graph, the developed pressure sensor was able to detect even the deep inspiration of the wearer. When the wearer jumped and air-punched at the same time, the sensor detected the pressure change produced by the movement between the karate body protector and the human body.

## 4. Conclusions

In this study, a highly sensitive textile pressure sensor was developed using piezoresistive film, conductive fabric as well as different bonding materials. The aim was to detect punches during sport karate; therefore, the best-built sensor was placed inside a karate body protector and tested. 

In order to develop the textile pressure sensor, small-scale sensors were created with different joining methods, namely, ultrasonic welding, hot press welding and oven curing, using TN and TW bonding materials. 

The sensors developed using the hot press welding technique were nonfunctional due to the possibility of silver nanoparticles being embedded into the piezoresistive film;Functional sensors were obtained by joining the layers at four sides using the ultrasonic welding technique as well as by using thermoplastic net (TN) and thermoplastic web (TW) materials. Due to the bonding materials’ uniform surface structure, TN gave the best results as a pressure sensor.

After using different methods and determining the best pressure sensor, the sensors were produced at a larger size to be placed inside the karate body protector. 

The sensor prepared using TN showed nonuniform pressure sensing ability, from which it can be concluded that, at large dimensions, nonuniform bonding may occur in the curing process.The sensor prepared using the double-sided film that was laser-cut to create adjacent circles of 15 mm in diameter also showed variations in output voltage in different measurement positions (po1 to po5) in the cyclic tests.Similar voltage outputs were obtained in the tests of the sensor produced using 2 mm diameters of adjacent circles cut into double-sided film in different measurement positions.The variability of the sensor responses at different positions may be explained by construction issues, but also by nonuniform resistances of the piezoelectric film used.

One of the large pressure sensors was tested in three ways, namely, placing the sensor on the wall, placing it on a dummy and placing it on a person.

Regarding the data obtained, it was clear that the sensor was able to detect every punch, even the light punches, and, as a result, it can be used in sport karate.

Future work should focus on establishing the exact causes of the variability of the sensors’ responses and designing methods of calibration for the sensors, both in static as well as in dynamic conditions. Another point of study is the selection of the ideal bonding material, which provides the highest delamination forces and the most uniform ratio of contact/noncontact areas over the surface of the film.

## Figures and Tables

**Figure 1 sensors-23-06524-f001:**
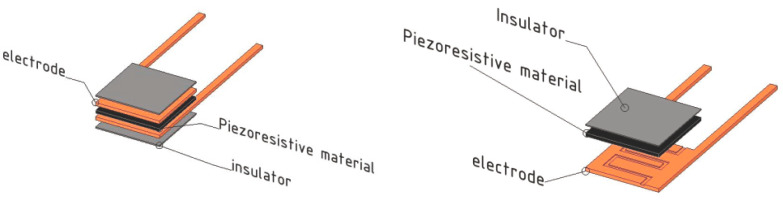
Sandwich (**left**) and one-layer (**right**) piezoresistive pressure sensor constructions [21].

**Figure 2 sensors-23-06524-f002:**
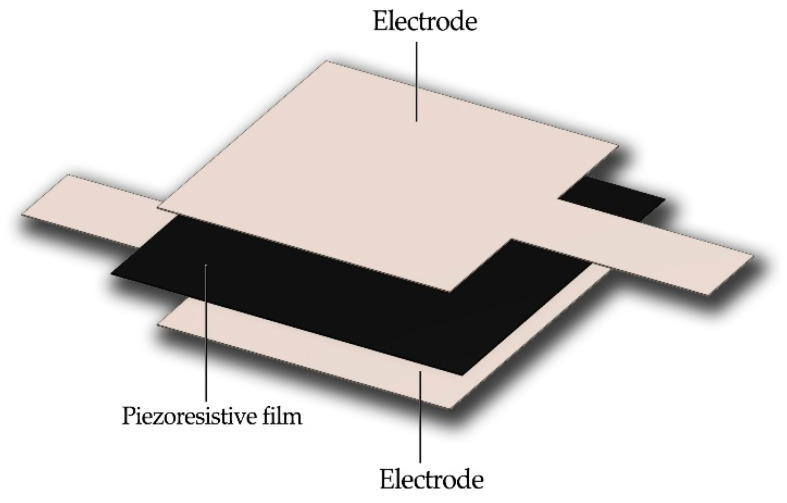
The construction of the pressure sensors.

**Figure 3 sensors-23-06524-f003:**
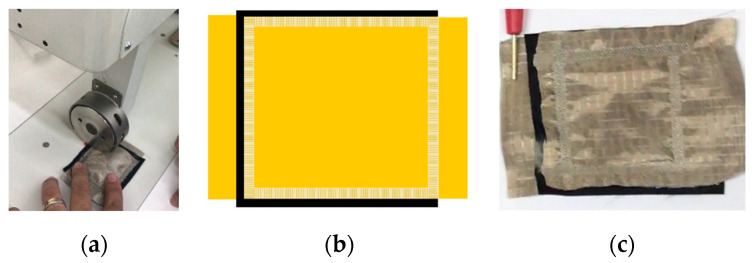
The joining process using ultrasonic welding (**a**), the construction of the sensor (**b**) and the generated sensor (**c**).

**Figure 4 sensors-23-06524-f004:**
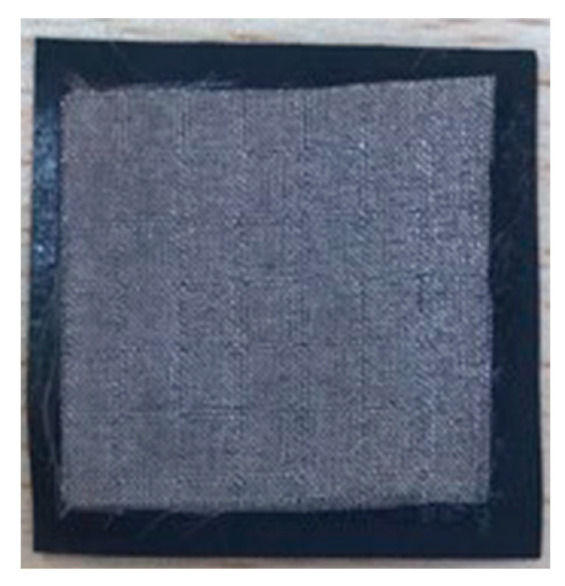
The sensor prepared using hot press welding.

**Figure 5 sensors-23-06524-f005:**
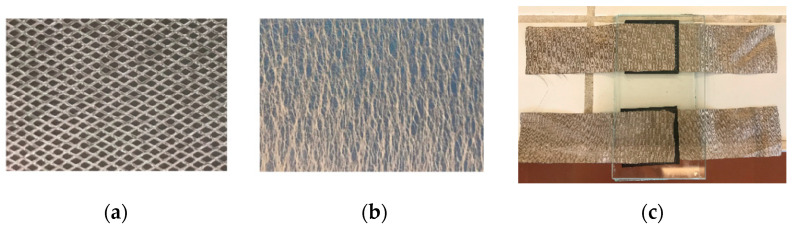
Thermoplastic bonding materials; (**a**) thermoplastic net (TN), (**b**) thermoplastic web (TW), (**c**) pressure sensor layout.

**Figure 6 sensors-23-06524-f006:**
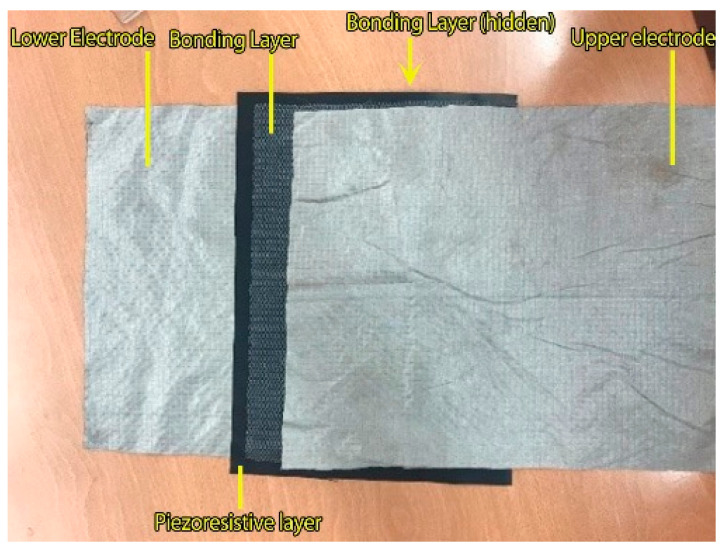
The large-scale sensor prepared using bonding material.

**Figure 7 sensors-23-06524-f007:**
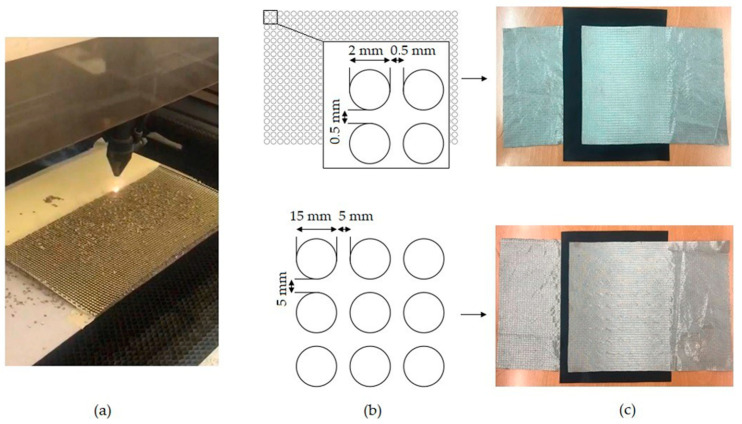
The laser cutting process of double-sided films (**a**), their design (**b**) and the obtained sensors (**c**).

**Figure 9 sensors-23-06524-f009:**
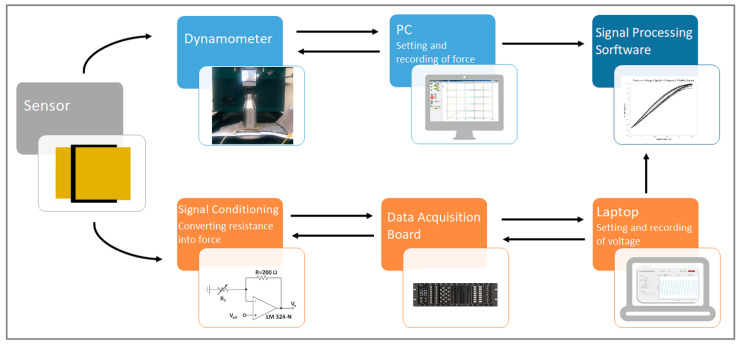
The measurement process.

**Figure 10 sensors-23-06524-f010:**
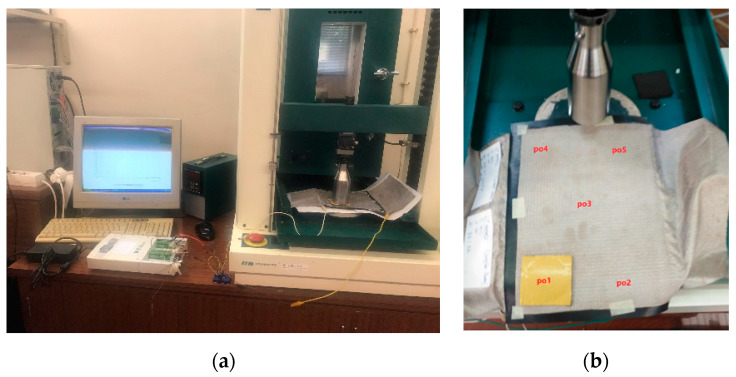
Setup for compression test (**a**); testing areas (**b**).

**Figure 11 sensors-23-06524-f011:**
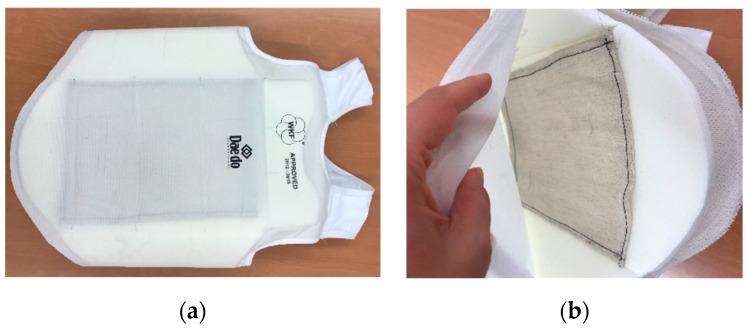
The body protector (**a**) unstitched and (**b**) the pocket sewn inside.

**Figure 12 sensors-23-06524-f012:**
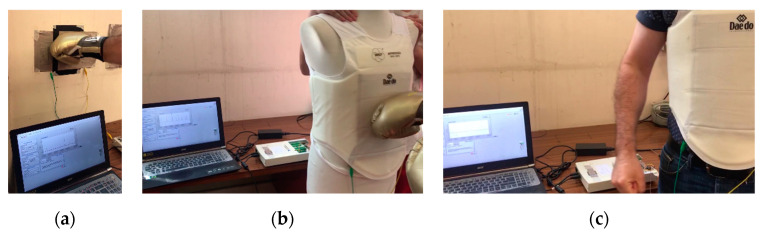
Real practice tests: (**a**) on the wall, (**b**) on the dummy, (**c**) on a person.

**Figure 13 sensors-23-06524-f013:**
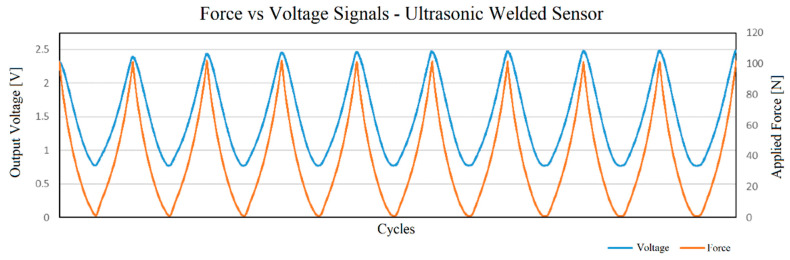
Waveforms of applied force and voltage signal acquired.

**Figure 14 sensors-23-06524-f014:**
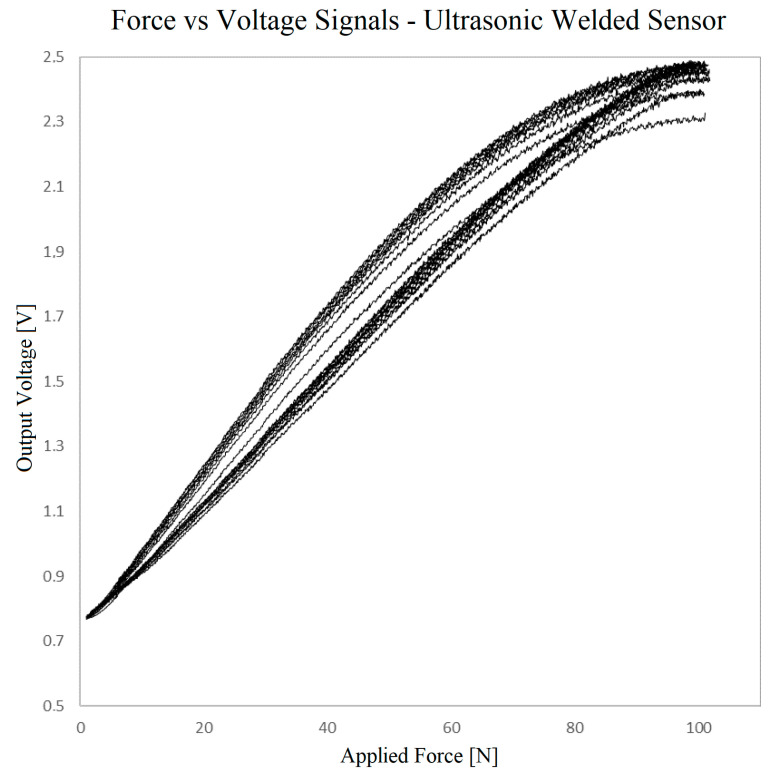
Voltage vs. force output for the sensor prepared using ultrasonic welding.

**Figure 15 sensors-23-06524-f015:**
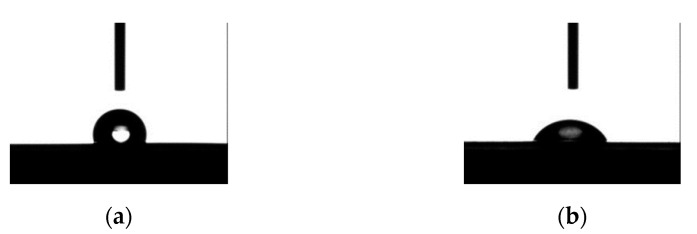
Contact angle (**a**) before plasma treatment, (**b**) after plasma treatment.

**Figure 16 sensors-23-06524-f016:**
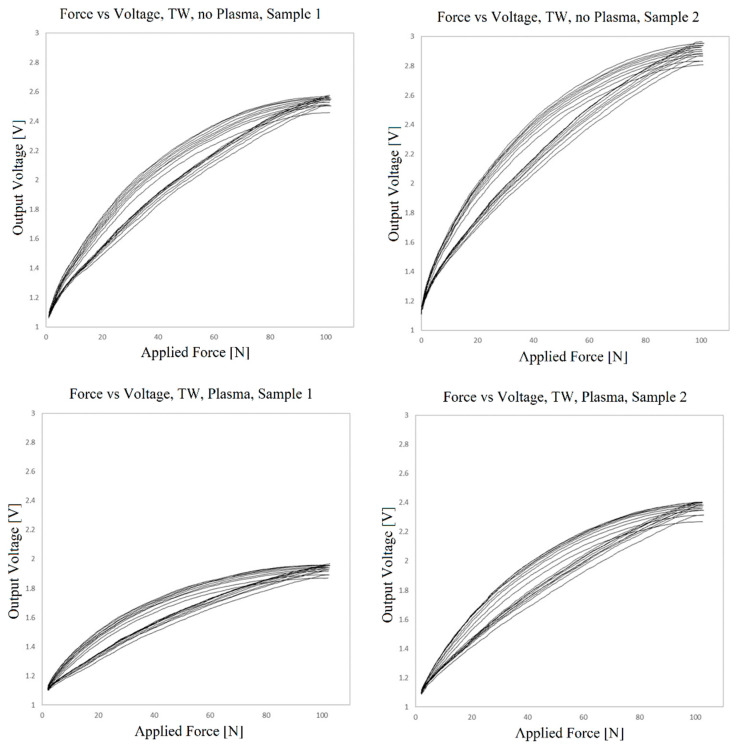
Voltage vs. force outputs for two samples of the sensor prepared using TW, with and without plasma treatment.

**Figure 17 sensors-23-06524-f017:**
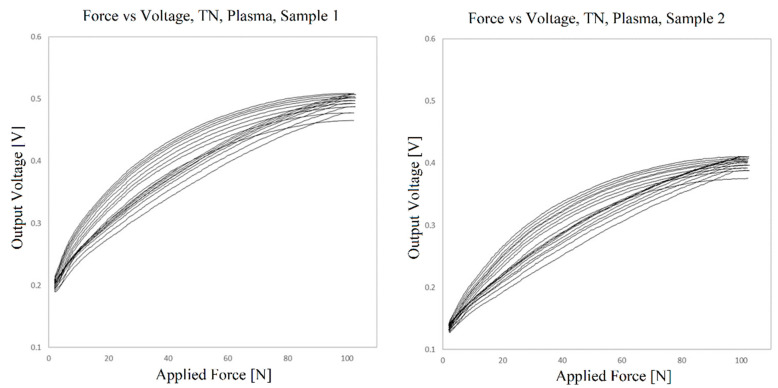
Voltage vs. force outputs for two samples of the sensor prepared using TN.

**Figure 18 sensors-23-06524-f018:**
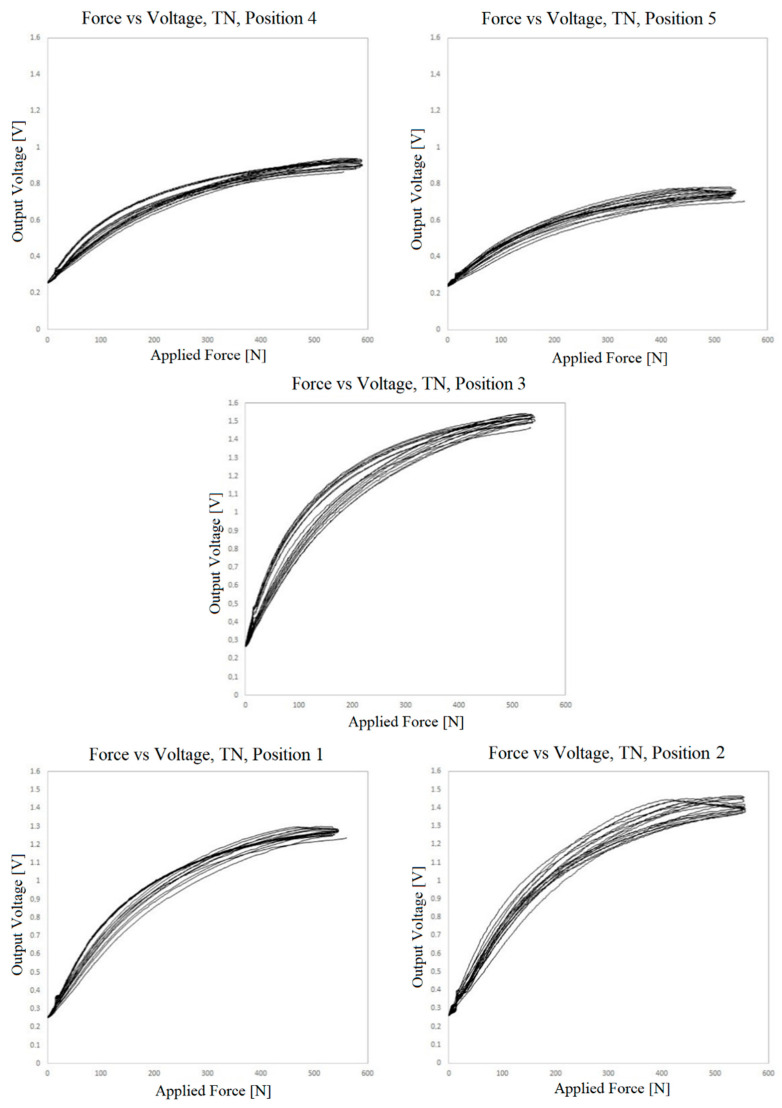
Voltage vs. force outputs at 5 positions for the sensor prepared with TN bonding material.

**Figure 19 sensors-23-06524-f019:**
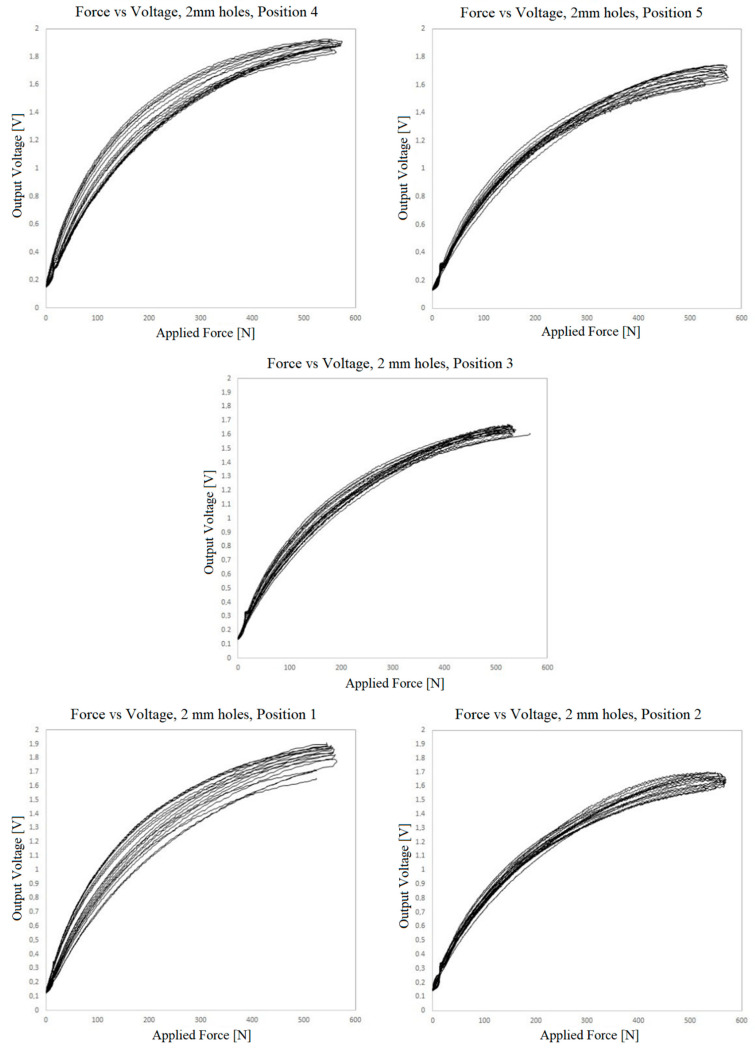
Voltage vs. force outputs at 5 positions for the sensor prepared with double-sided adhesive film and 2 mm holes.

**Figure 20 sensors-23-06524-f020:**
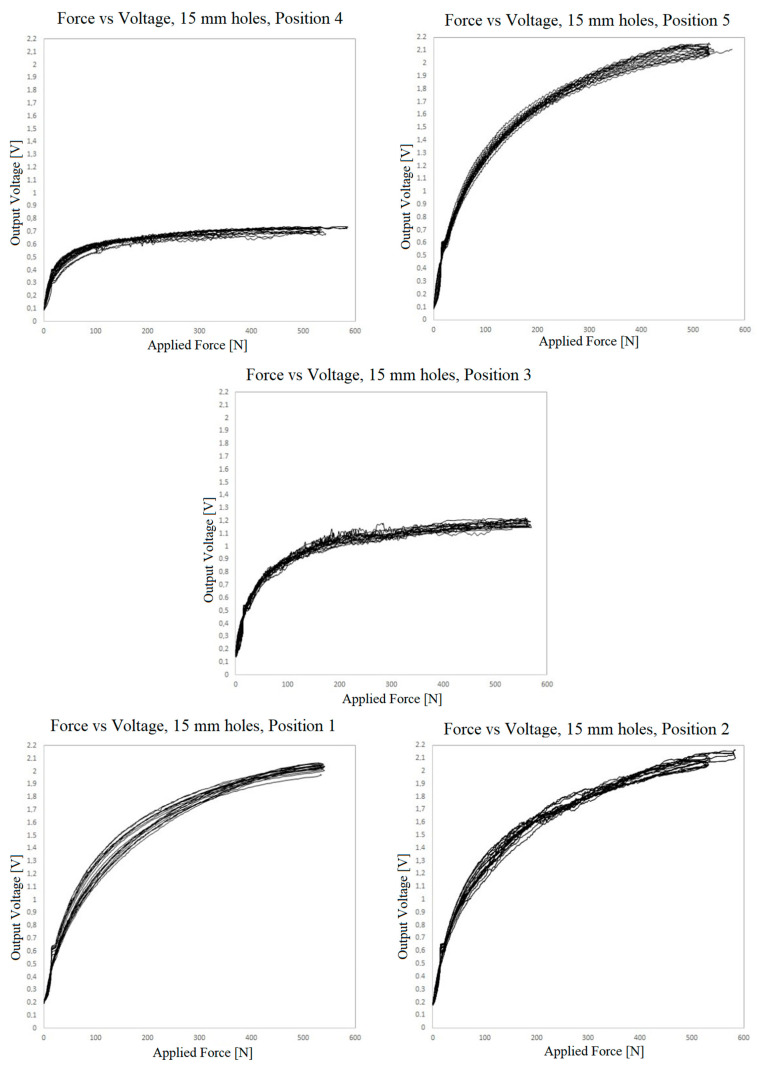
Voltage vs. force outputs at 5 positions for the sensor prepared with double-sided adhesive film and 15 mm holes.

**Figure 21 sensors-23-06524-f021:**
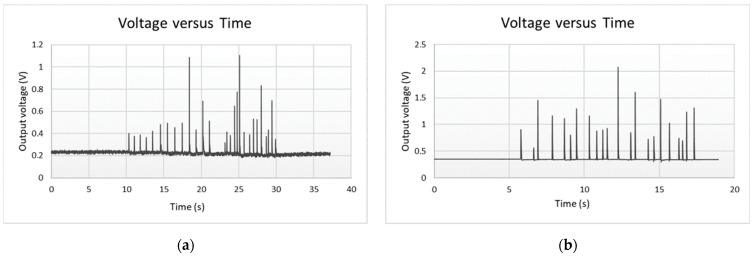
Real practice test results (**a**) on the wall on the wall, (**b**) on the dummy.

**Figure 22 sensors-23-06524-f022:**
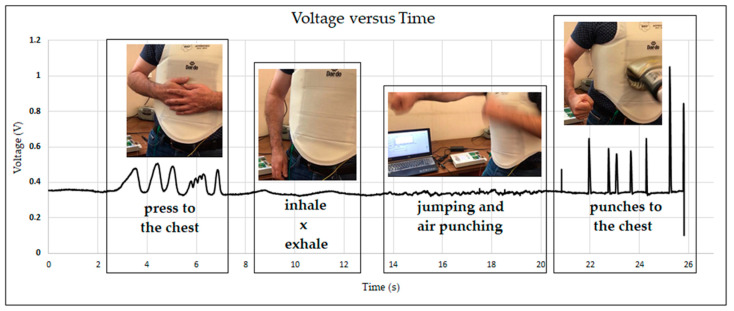
The pressure sensor-integrated karate body protector real practice test’s voltage by time data.

**Table 1 sensors-23-06524-t001:** Specifications of conductive polyethylene film and conductive fabric.

Specification	Conductive Polyethylene Film	Conductive Fabric
Description	Carbon-loaded PE	Silver-plated polyamide fabric (RS)
Plating	-	99% Pure silver
Surface Resistivity	±50,000 Ω/sq	Average < 0.3 Ω/sq
Total Thickness	0.1 mm	0.090 mm ± 12%
Weight	-	43 g/m² ± 10%
Temperature Range	up to 115 °C	−30 °C to 90 °C
Density	1 g/cm^3^	-

## Data Availability

Not applicable.

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
