# Peer review of "Flexible Pressure Sensors for Integration into Karate Body Protector"

_sensors, 2023, doi:10.3390/s23146524_

Round 1

Reviewer 1 Report

The paper focused on developing textile-based pressure sensors using piezoresistive film under different bonding materials and methods. Large scale sensors were manufactured to be placed inside the karate body protector and charcaterised. During these tests the body protector was placed on a dummy as well as on a person. The results showed that the piezoresistive textile-based pressure sensor produced is able to detect and quantify the impact of even light punches. It is interesting and innovative, and can be accepted after following revisions:

1) In figure 6, the name of all layers should be listed? and why there are four layers?

2)In figure 12, the environmental humidity will have effect on the piezoresistive charge, how do you consider it?

3)In figure 10, how do you study stability of voltage signal?

English grammar should be well checked.

Reviewer 2 Report

This study deals with how to develop a highly sensitive textile pressure sensor using piezoresistive films, conductive fabrics, and different adhesive materials. The authors used this technology to detect punches in karate sports. However, I think that if this technique is feasible, it should have more applications.

In addition, the authors have several areas for improvement

·       In lines 37-55, it is not necessary to introduce “World Karate Federation”. It is not relevant with the study.

·       In lines 65-71, the two paragraphs could be combined into one.

·   In section of 3.1.3, the authors mention that plasma treatment. But the authors don’t clearly tell the readers whether plasma treatment is good or not. First, it is based on a subjective analysis without any reference. Second, by examining the data in Fig. 16, the output voltage is smaller in the sensor with plasma treatment. That means that it becomes less sensitive. If this is the case, the authors need to clarify why they mention plasma treatment here.

·      In line 345, is it 0,1V or 0.1V? Same thing happens in all the plots.

·   For Fig. 16, it is better to plot data for samples with and without plasma treatment together. This is easier for the readers to figure out how plasma treatment works. It is same for Fig. 17-20, where it is better to plot the lines together to compare the differences.

·       In the conclusion section, the authors mention that there are differences in the response of the sensor at different locations. Can the authors provide a way to address this issue?

In summary, I think this paper needs to be improved before it can be accepted.

The authors need to highlight what they want to tell the readers. In this case, it is hard for me to tell whether plasma treatment is good or not.

Reviewer 3 Report

In this paper (sensors-2482333), the authors proposed a flexible textile pressure sensor for karate body protector. The application object is interesting, and the strategy and results are basically acceptable, however, there are some problems in the writing, experiment, results and disscussion. Some revisions need to be addressed before possible publication.

1. Abstract: It is recommended to provide sensor performance parameter data.

2. Merge the first and second parts (1. Introduction and objectives and 2. State-of-the-art) into an introduction to enhance logic.

3. It is recommended to delete Figure 1. If it is necessary to retain it, copyright permission is required.

4. Introduction: We know that flexible fabrics and cellulose paper pressure sensors have been widely reported. What are the current problems faced by flexible pressure sensors? Suggest analyzing and discussing the current research status of flexible pressure sensors (may refer to Sensors 2023, 23(5), 2443; J. Mater. Chem. C, 2023, 11, 5585–5600) to highlight innovation of this work.

5. Materials and Methods: Provide material information. For example, specifications, models, and characterization results (SEM, TEM) of carbon (Linqstat; Caplinq). In addition, daily carbon ink can also be used to prepare flexible pressure sensors. What are the advantages of this carbon compared to carbon ink used for daily writing? Suggest discussion.

6. The response and recovery times of the sensor need to be provided.

7. It is recommended to combine characterizations (such as optical photos and dynamic videos) to analyze the working mechanism of the sensor.

8. The author used 22 pictures to tell the work, which is obviously unreasonable. 6-8 pictures are suitable, and the rest are placed in the supporting material.

9. Suggest improving image quality, such as line width and font size.

10. Reference: Most of the references are outdated (before 2018), and flexible pressure sensors have developed rapidly. It is recommended to cite work from the past three years. In addition, Check the format of references. Check the completeness of reference information and abbreviations of journal names.

11. Check English writing.

Minor editing of English language required.

Round 2

Reviewer 2 Report

In the new version, the authors address all the questions I asked before. This paper can be accepted in present form.

Reviewer 3 Report

I carefully checked the response and revised manuscript. Concerns of  reviewer have been considered and addressed properly, and publication is recommended.

In addition, in the proof stage, it is necessary to correct the reference format, such as abbreviating the journal name.